# MOSS-CHATV: REINFORCEMENT LEARNING WITH PROCESS REASONING REWARD FOR VIDEO TEMPORAL REASONING

## ABSTRACT

Video reasoning has emerged as a critical capability for multimodal large language models (MLLMs), requiring models to move beyond static perception toward coherent understanding of temporal dynamics in complex scenes. Yet existing MLLMs often exhibit **process inconsistency**, where intermediate reasoning drifts from video dynamics even when the final answer is correct, undermining interpretability and robustness. To address this issue, we introduce **MOSS-ChatV**, a reinforcement learning framework with **a Dynamic Time Warping (DTW)–based process reward**. This rule-based reward aligns reasoning traces with temporally grounded references, enabling efficient process supervision without auxiliary reward models. We further identify dynamic state prediction as a key measure of video reasoning and construct **MOSS-Video**, a benchmark with annotated reasoning traces, where the training split is used to fine-tune MOSS-ChatV and the held-out split is reserved for evaluation. MOSS-ChatV achieves 87.2% on the MOSS-Video (test) and improves performance on general video benchmarks such as MVBench and MMVU. The framework consistently yields gains across different architectures, including Qwen2.5-VL and Phi2, confirming its broad applicability. Evaluations with GPT-4o-as-judge further show that MOSS-ChatV produces more consistent and stable reasoning traces.

## 1 INTRODUCTION

Multimodal Large Language Models (MLLMs) have shown remarkable progress in vision–language tasks such as image captioning, visual question answering, and video description (Cheng et al., 2024; Zhang et al., 2025a; Liang et al., 2024; Caffagni et al., 2024). Extending these advances from images to videos has attracted great attention, as videos contain richer temporal and causal information. However, video reasoning—requiring models to connect visual observations with temporal dynamics and causal dependencies—remains particularly challenging for current MLLMs.

Existing Video-MLLMs are predominantly trained through supervised fine-tuning on large-scale video–text pairs (Li et al., 2024a). While effective for basic understanding, this paradigm leaves models weak in reasoning-intensive tasks. A fundamental issue is the scarcity of datasets that provide fine-grained temporal reasoning supervision. Yet, videos inherently encode dense supervisory signals in their temporal evolution. The core challenge lies in exploiting these temporal signals to strengthen reasoning: models must not only recognize the present state but also infer future trajectories from context and world knowledge. Prior work such as VoT (Fei et al., 2024) has shown the close coupling between video prediction and reasoning, underscoring that temporal state prediction can serve as a proxy for reasoning ability. To operationalize this insight, we construct **MOSS-Video**, a dataset for video state prediction with annotated reasoning traces. The dataset is partitioned into training and test splits, enabling process-supervised learning while ensuring held-out evaluation.

Reinforcement learning (RL) offers a promising path for strengthening reasoning in MLLMs. However, recent studies (*e.g.,* , Video-UTR (Yu et al., 2025)) reveal a "temporal hacking" problem, where models bypass temporal reasoning and directly guess outcomes. This highlights the necessity of explicit process-level supervision. RL with process feedback has proven effective in domains such as mathematics and code generation (Shao et al., 2024; Ye et al., 2025). Motivated by this, we design

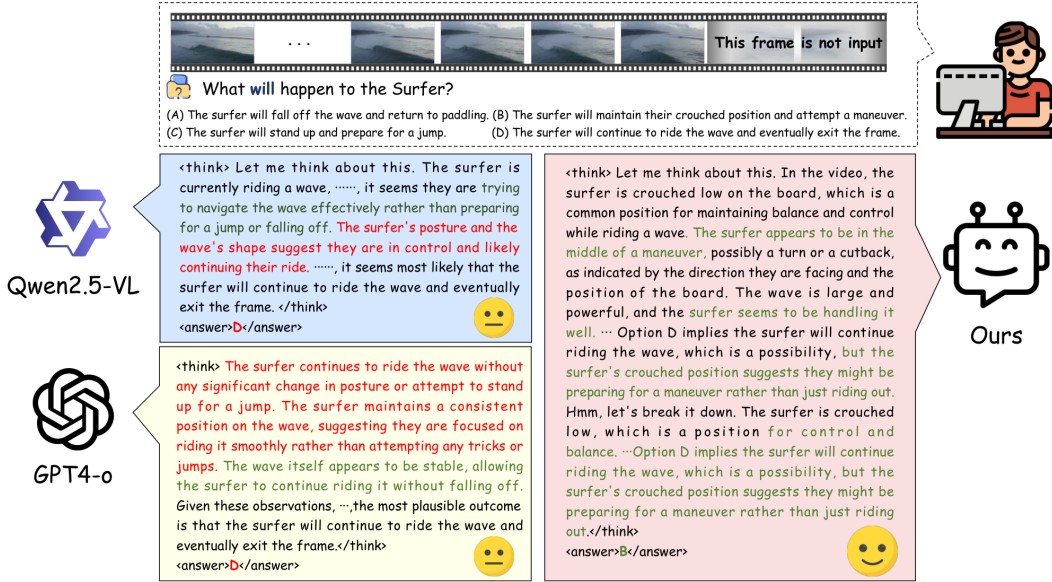

Figure 1: Illustration of the responses across different models on the video state prediction task, where **green**text indicates correctly reasoned key points and **red** text denotes reasoning errors. Comparative analysis reveals that MOSS-ChatV captures more fine-grained states (*e.g.*, the surfer's crouched position) compared to other models. Crucially, it accurately extrapolates this state (preparing for a maneuver), thereby achieving more coherent and correct reasoning.

a rule-based **Process Reasoning Reward (PRR)** for video reasoning. Specifically, we employ a two-stage "split-align" strategy: (1) decomposing reasoning traces into sequential substeps, and (2) aligning generated and reference processes via subsequence Dynamic Time Warping (DTW). The resulting alignment distance provides a reward signal that supervises temporal coherence without the need for a learned reward model. Leveraging PRR together with the MOSS-Video training split, we fine-tune **MOSS-ChatV** using GRPO (DeepSeek-AI et al., 2025), as illustrated in Figure 2.

Extensive experiments validate the effectiveness of our approach. See figure 1 for the case demonstrations. **MOSS-ChatV** achieves 87.2% accuracy on the MOSS-Video test set, surpassing strong closed-source baselines such as GPT-4o. It also improves general video understanding, reaching 67.6% on MVBench (Li et al., 2024b), and performs competitively on real-time benchmarks such as RTVBench (Xun et al., 2025). Moreover, the framework consistently boosts reasoning quality across architectures including Qwen2.5-VL and TinyLLaVA-Video. Automatic evaluation with GPT-4o as a judge further shows that MOSS-ChatV produces more consistent and stable reasoning traces. Our main contributions are as follows:

- We construct **MOSS-Video**, a video state prediction dataset with reasoning annotations, split into training and test partitions for process-supervised reinforcement learning and held-out evaluation.

- We propose a rule-based **Process Reasoning Reward (PRR)** based on subsequence DTW and integrate it into a reinforcement learning framework, **MOSS-ChatV**, trained with GRPO. This design enables efficient temporal alignment and process supervision without training additional reward models.

- Through extensive experiments, we demonstrate that MOSS-ChatV achieves state-of-the-art performance on the MOSS-Video (test), improves general video understanding benchmarks such as MVBench and MMVU, and yields consistent gains across different architectures including Qwen2.5-VL and TinyLLaVA-Video.

## 2 PRELIMINARY

### 2.1 VIDEO STATE PREDICTION AND REASONING

We consider video state prediction as follows: given a video $V$ and a query $q$ specifying a target object, the model must (i) identify the object, (ii) infer its current or imminent state, and (iii) provide a temporally grounded explanation. An illustrative example is shown in Figure 1. VoT (Fei et al., 2024) demonstrates that decomposing the task via Chain-of-Thought (CoT)—including task definition, object recognition/tracking, behavior analysis, answer ranking, and verification—yields a human-like reasoning path and highlights the tight coupling between prediction and reasoning. Different from the prompt-based paradigm in VoT, our approach learns this capability via reinforcement learning with a process-level reward, integrating temporal reasoning into the model's latent space to enable end-to-end reasoning and prediction.

### 2.2 GROUP RELATIVE POLICY OPTIMIZATION (GRPO)

Recent work (DeepSeek-R1) (DeepSeek-AI et al., 2025) introduced Group Relative Policy Optimization (GRPO), which has spurred effective adaptations for multimodal LLMs (Feng et al., 2025; Li et al., 2025; Wang et al., 2025b; Zhang et al., 2025b). At a high level, for each input, GRPO samples a group of $G$ candidate responses from the current policy $\pi_\theta$, compares their relative performance via a scalar reward, and updates the policy without learning a value function. We adopt GRPO as our optimization backbone due to its simplicity and strong empirical stability.

**Notation.** For one input, let the sampled response set be $\mathcal{O} = \{o_i\}_{i=1}^{G}$ with corresponding scalar rewards $\{\mathcal{R}_i\}_{i=1}^{G}$. GRPO computes a standardized advantage for each response:

$$A_i \;=\; \frac{\mathcal{R}_i - \mu}{\sigma}, \qquad \mu \;=\; \text{mean}\big(\{\mathcal{R}_i\}_{i=1}^{G}\big), \quad \sigma \;=\; \text{std}\big(\{\mathcal{R}_i\}_{i=1}^{G}\big). \tag{1}$$

The learning objective encourages higher-advantage responses under importance weighting, while regularizing the policy against a fixed reference policy $\pi_{\text{ref}}$:

$$\mathcal{L}_{\text{GRPO}}(\theta) \;=\; \mathbb{E}_{\mathbf{o}\sim(\pi_\theta^{\text{old}})} \left[ \frac{1}{G} \sum_{i=1}^{G} \min\left( \frac{\pi_\theta(o_i)}{\pi_\theta^{\text{old}}(o_i)} A_i, \; \text{clip}\left( \frac{\pi_\theta(o_i)}{\pi_\theta^{\text{old}}(o_i)}, 1-\epsilon, 1+\epsilon \right) A_i \right) \right] \\ - \; \beta \, D_{\text{KL}}\big(\pi_\theta \,\big\|\, \pi_{\text{ref}}\big) \tag{2}$$

Here $\pi_\theta^{\text{old}}$ denotes the behavior policy used for sampling the group, $\epsilon$ denotes the range of the clip operation, and $D_{\text{KL}}(\cdot\|\cdot)$ is the Kullback–Leibler divergence. The importance ratio reweights each response $o_i$ to correct for the sampling distribution, while the KL term (scaled by $\beta > 0$) controls policy drift.

**Accuracy Reward.** For multiple-choice or short-answer settings, a binary accuracy signal provides a simple yet effective supervision:

$$\mathcal{R}_{\text{acc}}(a_{\text{model}}, a_{\text{gt}}) \;=\; \begin{cases} 1, & \text{if } a_{\text{model}} = a_{\text{gt}}, \\ 0, & \text{otherwise}. \end{cases} \tag{3}$$

**Format Reward.** In many applications, outputs must follow a specified schema (*e.g.,* , `<think>...</think><answer>...</answer>`) to expose intermediate reasoning. Let $o_{\text{model}}$ denote the full model output and $\mathcal{F}$ the required format:

$$\mathcal{R}_{\text{fmt}}(o_{\text{model}}, \mathcal{F}) \;=\; \begin{cases} 1, & \text{if } o_{\text{model}} \text{ adheres to } \mathcal{F}, \\ 0, & \text{otherwise}. \end{cases} \tag{4}$$

Accuracy and format rewards are effective foundations for RL fine-tuning, but they do not explicitly supervise temporal logic. In our method (Section 3), we therefore introduce a process-level reward to align intermediate reasoning with reference temporal processes, complementing $\mathcal{R}_{\text{acc}}$ and $\mathcal{R}_{\text{fmt}}$ within the GRPO framework. Algorithm 1 summarizes the overall optimization steps.

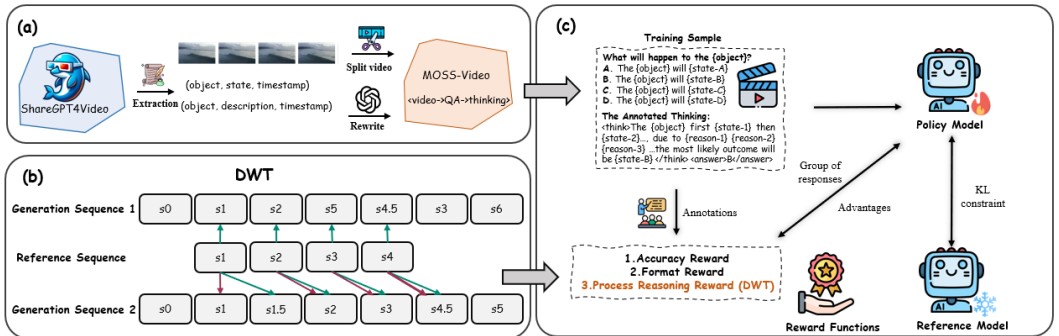

Figure 2: Overall training pipeline of MOSS-ChatV. (a) Construction of the MOSS-Video dataset from ShareGPT4Video with multi-level temporal annotations, where future states are masked as prediction targets. (b) Subsequence DTW alignment: green dashed lines denote strict sequential matching, while red solid lines allow jumps (*jump step*=2) to reduce cumulative distance. (c) GRPO workflow integrating accuracy, format, and process rewards.

## 3 PROCESS REASONING REWARD

Addressing the limitations of conventional rewards in guiding complex temporal reasoning, we introduce a Process Reasoning Reward (PRR), denoted as $\mathcal{R}_{\text{proc}}$. This reward leverages reference annotations embodying an ideal 'gold standard' reasoning process. Crucially, this mechanism achieves nuanced process supervision by effectively leveraging efficient, robust algorithms, avoiding the need for potentially complex or computationally expensive large model-based evaluators.

**Reasoning Step Serialization** The first step is segmentation for reasoning texts. The model's intermediate reasoning (*e.g.*, content within <think>...</think> tags) and the reference counterpart are segmented into sequences of textual steps using NLP tools (*e.g.*, nltk library). Though not affecting overall temporal information, this segmentation enables finer-grained analysis in the next step by splitting long texts into sequences.

Let $T_{gen}$ represent the intermediate reasoning content generated by the model, and $T_{ref}$ represent the reference reasoning content. These are segmented into sequences of textual steps using NLP tools (denoted as $\mathcal{N}$):

$$Seq_{gen} = \{g_1, \ldots, g_m\} = \mathcal{N}(T_{gen}) \tag{6}$$
$$Seq_{ref} = \{r_1, \ldots, r_n\} = \mathcal{N}(T_{ref}) \tag{7}$$

**Temporal Alignment via Subsequence DTW** For the second step, We employ Subsequence Dynamic Time Warping (SDTW), detailed in Algorithm 2, a highly efficient dynamic programming algorithm, to quantify the alignment between two sequences with different lengths. SDTW optimally identifies the best-matching subsequence within the model's reasoning sequence ($Seq_{gen}$) that corresponds to the entire reference sequence ($Seq_{ref}$), by minimizing a cumulative distance. This cumulative distance, minimized by SDTW, is built upon the pairwise distances $d(g_j, r_i)$ between an individual generated step $g_j \in Seq_{gen}$ and the annotated reference step $r_i \in Seq_{ref}$. To define $d(g_j, r_i)$, our goal is to comprehensively yet efficiently measure the textual similarity between these steps. This is achieved by leveraging several rule-based ROUGE scores. We use ROUGE-1 and ROUGE-2 to capture n-gram overlap between $g_j$ and $r_i$. To evaluate sequence-level structural similarity, ROUGE-L is used for preserving the sentence-internal logical order within each step.

The distance $d(g_j, r_i)$ is then formally defined as one minus the average of average of these ROUGE scores:

$$\text{ROUGE}_{\text{avg}}(g_j, r_i) = \frac{\text{ROUGE-1}(g_j, r_i) + \text{ROUGE-2}(g_j, r_i) + \text{ROUGE-L}(g_j, r_i)}{3} \tag{8}$$

$$d(g_j, r_i) = 1 - \text{ROUGE}_{\text{avg}}(g_j, r_i) \tag{9}$$

The minimum cumulative distance is then defined as:

$$D_{sdtw} = SUBSEQUENCE\_DTW(D) \tag{10}$$

---

**Algorithm 1** GRPO with Process Reasoning Reward (PRR) for one training sample

---

**Require:** Training sample $(V, Q, a_{\text{gt}}, T_{\text{ref}})$; Policy model $M_{\text{policy}}(\pi_\theta)$; Reference model $M_{\text{ref}}(\pi_{\text{ref}})$; Format $\mathcal{F}$

1: Sample $G$ candidate outputs from policy: $\mathcal{O} = \{o_i = (T_{\text{gen}}, a_{\text{model}})\}_{i=1}^G$
2: Initialize reward list $\mathcal{R} = [\,]$
3: **for** each $o_i \in \mathcal{O}$ **do**
4:      Segment reference: $Seq_{\text{ref}} = \{r_1, \ldots, r_n\} \leftarrow \mathcal{N}(T_{\text{ref}})$
5:      Segment generation: $Seq_{\text{gen}} = \{g_1, \ldots, g_m\} \leftarrow \mathcal{N}(T_{\text{gen}})$
6:      Build distance matrix $\mathbf{D} \in \mathbb{R}^{m \times n}$ with

$$D_{j,k} = 1 - \text{ROUGE}_{\text{avg}}(g_j, r_k), \quad j \in [1, m],\ k \in [1, n].$$

7:      Compute $D_{\text{sdtw}} \leftarrow \text{SUBSEQUENCE\_DTW}(\mathbf{D})$
8:      Process reward: $\mathcal{R}_{\text{proc}} = \exp(-\alpha \cdot D_{\text{sdtw}})$
9:      Total reward:
$$\mathcal{R}_i = \mathcal{R}_{\text{acc}}(a_{\text{model}}, a_{\text{gt}}) + \mathcal{R}_{\text{fmt}}(o_i, \mathcal{F}) + \mathcal{R}_{\text{proc}}$$

10:     Append $\mathcal{R}_i$ to $\mathcal{R}$
11: **end for**
12: Standardize advantages:

$$A_i = \frac{\mathcal{R}_i - \mu}{\sigma}, \quad \mu = \text{mean}(\mathcal{R}),\ \sigma = \text{std}(\mathcal{R})$$

13: Compute GRPO objective with clipping:

$$\mathcal{L}_{\text{GRPO}}(\theta) = \mathbb{E}_{\mathbf{o} \sim \pi_\theta^{\text{old}}} \left[ \frac{1}{G} \sum_{i=1}^{G} \min \left( \frac{\pi_\theta(o_i)}{\pi_\theta^{\text{old}}(o_i)} A_i,\ \text{clip}\left( \frac{\pi_\theta(o_i)}{\pi_\theta^{\text{old}}(o_i)},\ 1 - \epsilon,\ 1 + \epsilon \right) A_i \right) \right] \quad (5)$$
$$- \beta\, D_{\text{KL}}(\pi_\theta \,\|\, \pi_{\text{ref}}).$$

14: Update policy: $M_{\text{policy}}.\text{update}(\mathcal{L}_{\text{GRPO}})$

---

---

**Algorithm 2** Subsequence DTW

---

1: **function** SUBSEQUENCE\_DTW($\mathbf{D}, k_{\text{ref}}, k_{\text{target}}$) ▷ $\mathbf{D}$: Cost matrix ($n \times m$), $k_{\text{ref}}$: max reference jump, $k_{\text{target}}$: max target jump
2:      Initialize $\mathcal{P} \in \mathbb{R}^{(n+1) \times (m+1)}$ with $\mathcal{P}[0, j] \leftarrow 0$ for $j \in [0, m]$, $\mathcal{P}[i, 0] \leftarrow \infty$ for $i \in [1, n]$
3:      **for** $i \leftarrow 1$ to $n$ **do**
4:          **for** $j \leftarrow 1$ to $m$ **do**
5:              diag\_cost $\leftarrow \mathcal{P}[i-1, j-1]$         ▷ Match current points (diagonal move)
6:              up\_cost $\leftarrow \min_{1 \le k \le \min(k_{\text{ref}}, i)} \mathcal{P}[i-k, j]$    ▷ Skip $k$ points in reference sequence (vertical move)
7:              left\_cost $\leftarrow \min_{1 \le k \le \min(k_{\text{target}}, j)} \mathcal{P}[i, j-k]$    ▷ Skip $k$ points in target sequence (horizontal move)
8:              $\mathcal{P}[i, j] \leftarrow \mathbf{D}[i, j] + \min(\text{diag\_cost}, \text{up\_cost}, \text{left\_cost})$
9:          **end for**
10:     **end for**
11:     **return** $\min_{j \in [1, m]} \mathcal{P}[n, j]$        ▷ Shortest distance to any endpoint in target sequence
12: **end function**

---

We adopt Subsequence Dynamic Time Warping (SDTW) for its ability to align a reference reasoning path ($Seq_{ref}$) within potentially longer model-generated sequences ($Seq_{gen}$), enabling process supervision with explicit temporal signals. A key advantage is SDTW's compatibility with reinforcement learning: it avoids penalizing exploratory segments outside the optimal alignment while still rewarding correct paths. The algorithm provides tunable alignment strictness through parameters like *jump steps* (Algorithm 2, figure 2), permitting controlled tolerance for minor deviations in the reasoning trajectory. This balance of flexibility and precision makes SDTW ideal for guiding reasoning processes without stifling exploration.

**Distance-to-Reward Transformation** The final minimum cumulative distance $D_{sdtw}$ from SDTW is transformed into the reward value $\mathcal{R}_{\text{proc}}$ via a transformation function $\mathcal{T}$:

$$\mathcal{R}_{\text{proc}} = \mathcal{T}(D_{sdtw}) \tag{11}$$

$$\mathcal{T}(D_{sdtw}) = \exp(-\alpha \cdot D_{sdtw}) \tag{12}$$

where $\alpha > 0$ is a tunable hyperparameter that controls the sensitivity or decay rate of the reward with respect to the distance.

Then we can get the total reward $\mathcal{R}_{\text{total},i}$ for the $i$-th response in the sampled group of responses, by combining its specific process reward $\mathcal{R}_{\text{proc},i}$ with its accuracy $\mathcal{R}_{\text{acc},i}$ and format $\mathcal{R}_{\text{fmt},i}$:

$$\mathcal{R}_{\text{total},i} = \mathcal{R}_{\text{proc},i} + \mathcal{R}_{\text{acc},i} + \mathcal{R}_{\text{fmt},i} \tag{13}$$

This $\mathcal{R}_{\text{total},i}$ corresponds to the $\mathcal{R}_i$ used in the GRPO advantage calculation (Equation **??**) for the $i$-th response within the group $\{o_1, \ldots, o_G\}$.

Consequently, the resulting $\mathcal{R}_{proc}$ provides a computationally efficient yet powerful reward signal for reinforcement learning. It uniquely encourages temporal coherence in reasoning, validates the inclusion and ordering of essential logical steps, and maintains sensitivity to the relevance of generated content, thereby offering comprehensive guidance towards generating both accurate and logically sound reasoning processes.

**MOSS-Video Dataset** To support process-supervised reinforcement learning, we construct **MOSS-Video**, a large-scale video state prediction dataset derived from ShareGPT4Video (Chen et al., 2024). Each sample is annotated with object states and corresponding reasoning traces, enabling models to predict future states conditioned on visual context. The dataset is partitioned into a training split (11,654 samples, 1,218 unique videos) and a held-out test split (2,836 samples, 479 unique videos). Basic statistics are summarized in Table 1, including average video length and annotation span. Annotation pipelines and further details are provided in Appendix A.

Table 1: Comparison of MOSS-Video with representative video temporal reasoning datasets. Our dataset uniquely supports state prediction with explicit reasoning annotations.

| Dataset | #Samples | Avg. Video Len (s) | Understanding | Reasoning | Prediction |
|---|---|---|---|---|---|
| ViTiB (Zhang et al., 2023) | 1,382 | – | ✓ | ✓ | ✗ |
| NeXT-QA (Xiao et al., 2021) | 3,870 | 40 | ✓ | ✗ | ✗ |
| Video-R1-CoT-165K (Feng et al., 2025) | 116k | – | ✗ | ✓ | ✗ |
| MOSS-Video (train) | 11,654 | 27.73 | ✓ | ✓ | ✓ |
| MOSS-Video (test) | 2,836 | 28.21 | ✓ | ✓ | ✓ |

## 4 EXPERIMENT

We directly performed reinforcement fine-tuning on the Qwen2.5VL model, leveraging the training frameworks provided by Open-R1-Video (Wang & Peng, 2025) and Video-R1 (Feng et al., 2025), and utilizing the MOSS-Video train set. We selected a comprehensive suite of benchmarks for the holistic evaluation of MOSS-ChatV. This suite includes MVBench (Li et al., 2024b), TempCompass (Liu et al., 2024a), Video MME (Fu et al., 2024), RTV-Bench (Xun et al., 2025) and the MOSS-Video test set for our state prediction scenarios. These benchmarks collectively assess a wide range of video understanding capabilities, including temporal reasoning, action recognition, causal inference, and narrative comprehension. To demonstrate our method's generalizability, we further experiment on TinyLLaVA-Video (Zhang et al., 2025b), validating its effectiveness with a different language model (Phi2) and visual encoder (SigLIP). The aggregated evaluation results are presented in Table 2. Specific configurations for our evaluations included a sampling temperature of 0 to ensure deterministic outputs and an input video resolution of approximately 448x448 pixels. We tested our experiment on 4 NVIDIA A800 and trained MOSS-ChatV on 8 NVIDIA A800.

### 4.1 RESULTS

In Table 2, our MOSS-ChatV model achieves state-of-the-art performance on MVBench, VideoMME, RTVBench, and MOSS-Video test compared to baseline models, Qwen2.5-VL, and the same architecture model Video-R1. It also demonstrates improvements over the Qwen2.5-VL on TempCompass.

Table 2: Results of MOSS-ChatV and baselines on (a) general video understanding benchmarks and (b) video reasoning benchmarks. All results use 32-frame input setting. Our method consistently improves performance across both categories.

(a) General Benchmarks

| Model | # LLM | MVBench | VideoMME | TempCompass |
|---|---|---|---|---|
| Qwen2.5-VL Bai et al. (2025) | Qwen2.5-7B | 67.1 | 59.7 | 72.2 |
| LLaVA-OneVision Li et al. (2024a) | Qwen2-7B | 56.7 | 58.2 | – |
| TinyLLaVA-3B Zhang et al. (2025b) | Phi2-3B | 28.8 | 34.5 | 32.4 |
| TinyLLaVA-3B + PRR | Phi2-3B | 29.0 | 35.1 | 45.1 |
| Video-UTR Yu et al. (2025) | Qwen2-7B | 58.8 | 52.6 | 59.7 |
| VideoChat-R1 Li et al. (2025) | Qwen2.5-7B | 66.2 | 58.8 | 73.9 |
| VideoChat-R1-thinking Li et al. (2025) | Qwen2.5-7B | – | 58.3 | **75.0** |
| Video-R1 Feng et al. (2025) | Qwen2.5-7B | 63.9 | 59.3 | 73.2 |
| **MOSS-ChatV (ours)** | Qwen2.5-7B | **67.6** | **60.0** | 72.9 |

(b) Reasoning Benchmarks

| Model | RTV-Bench | MOSS-Video$_{test}$ | MMVU$_{mc}$ | VideoMMMU | VCR-B$_{mc}$ | VSI-B$_{mc}$ | VSI-B$_{reg}$ |
|---|---|---|---|---|---|---|---|
| Qwen2.5-VL | 32.8 | 67.0 | 60.0 | 48.1 | 33.7 | 35.3 | 24.3 |
| LLaVA-OneVision | 34.5 | 48.1 | – | – | – | – | – |
| TinyLLaVA-3B | – | 65.9 | 39.0 | – | – | – | – |
| TinyLLaVA-3B + PRR | – | 82.5 | 40.3 | – | – | – | – |
| Video-UTR | – | 58.9 | – | – | – | – | – |
| VideoChat-R1 | – | 70.8 | 62.7 | 50.0 | 34.5 | – | – |
| VideoChat-R1-thinking | – | 70.1 | 64.2 | 49.2 | 35.3 | **35.9** | 30.0 |
| Video-R1 | 46.5 | 73.3 | 64.8 | **52.3** | **38.4** | 30.8 | **39.7** |
| **MOSS-ChatV (ours)** | **46.6** | **86.6** | **66.2** | 50.2 | 35.3 | 35.2 | 28.2 |

The results from MOSS-Video test particularly indicate that reasoning capabilities contribute positively to video prediction tasks. Notably, while neither Qwen2.5-VL nor Video-R1 were trained on MOSS-Video Train data, but Video-R1 shows significant metric improvements, suggesting the benefits of reasoning.

We tested MOSS-Video using different input number of frames, result shown in Figure 3. The results demonstrate that increasing input frames enhances state prediction performance. MOSS-ChatV likely reaches peak accuracy with fewer frames due to its more efficient information extraction and reasoning capability.

It is worth emphasizing that our solution utilizes only a single-task dataset for video prediction, yet achieves performance gains across general video benchmarks. The improvements are especially pronounced on MVBench and VideoMME - both requiring complex reasoning - demonstrating that our approach effectively unlocks the model's latent potential. These

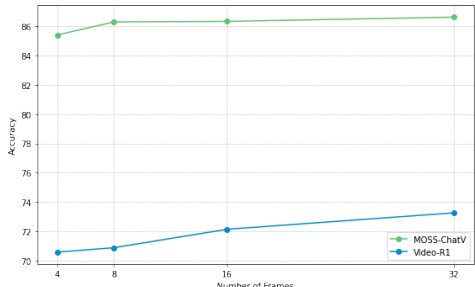

Figure 3: Performance impact of varying input frame counts.

results collectively provide evidence that video prediction tasks indeed enhance models' reasoning capabilities.

## 4.2 SDTW vs. DTW

When comparing reasoning sequences, traditional Naive Dynamic Time Warping (DTW) and Subsequence DTW exhibit distinct behaviors. Naive DTW attempts to achieve a complete match between two sequences through warping, which can lead to a single element being mapped to multiple elements. This significantly inflates the distance metric for sequences of unequal length, a characteristic we find undesirable as it unduly penalizes valid model explorations that extend beyond the shortest annotated reasoning chain. Our experiments, Figure 4, demonstrate that Naive DTW can induce a "reward hacking" phenomenon: when the model outputs very short reasoning, the

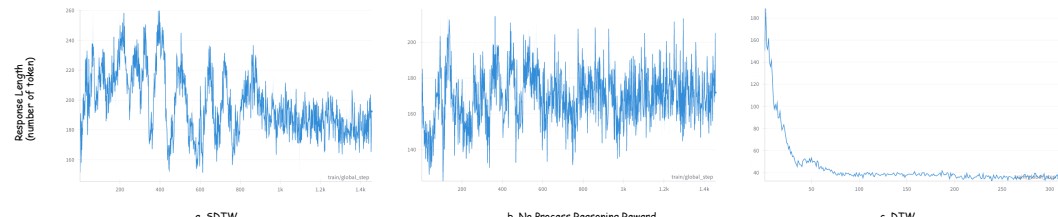

Figure 4: Figure (a) shows that with Subsequence DTW (SDTW), response lengths initially fluctuate due to exploration but gradually converge to a stable range. Figure (b) reports training without process supervision, where response lengths remain unstable. Figure (c) illustrates Naive DTW, which induces reward hacking: the model shortens its reasoning drastically to exploit the distance metric.

DTW distance is minimized, leading to trivial outputs like `<think>Based on the video content, the correct answer is A</think><answer>A</answer>`. This observation also highlights that treating annotated reasoning processes solely as an absolute "gold standard" for training offers limited benefits for model improvement. Consequently, our strategy positions annotated reasoning as a "minimal" gold standard. While ensuring the quality of reasoning, this approach avoids overly restricting the model's legitimate explorations beyond this baseline, thereby aiming to more comprehensively unlock and leverage the model's latent potential.

### 4.3 ABLATION STUDY

Table 3: Ablation Results

| Model | MVBench | VideoMME | MOSS-Video |
|---|---|---|---|
| Qwen2.5-VL-7B | 67.09 | 59.67 | 67.00 |
| Qwen2.5-VL-7B+SFT (MOSS-ChatV-SFT) | 65.12 $\downarrow_{1.97}$ | 55.24 $\downarrow_{4.43}$ | 71.44 $\uparrow_{4.44}$ |
| Qwen2.5-VL-7B+T-GRPO (Video-R1) | 63.90 $\downarrow_{3.19}$ | 59.30 $\downarrow_{0.37}$ | 73.26 $\uparrow_{6.26}$ |
| Qwen2.5-VL-7B+GRPO (MOSS-ChatV-no-PPR) | 65.23 $\downarrow_{1.86}$ | 55.30 $\downarrow_{4.37}$ | 84.17 $\uparrow_{17.17}$ |
| Qwen2.5-VL-7B+GRPO+Process Reasoning Reward (MOSS-ChatV) | **67.60** $\uparrow_{0.51}$ | **59.96** $\uparrow_{0.29}$ | **86.62** $\uparrow_{19.62}$ |

We conduct ablation experiments using the MOSS-Video Train dataset, comparing three variants: MOSS-ChatV , MOSS-ChatV-no-PRR (MOSS-ChatV without process supervision), and supervised fine-tuned MOSS-ChatV-SFT. The results (see Table 3) demonstrate that the complete MOSS-ChatV achieves superior performance across all benchmarks. The absence of process supervision in MOSS-ChatV-no-PRR leads to degraded temporal reasoning performance, confirming the importance of alignment signals for video understanding. Notably, even without temporal supervision, MOSS-ChatV-no-PRR outperforms MOSS-ChatV-SFT, highlighting the advantages of reinforcement learning over pure supervised training for video reasoning tasks.

### 4.4 MLLM AS A JUDGE FOR REASONING QUALITY EVALUATION

Table 4: MLLM as a judge for evaluating the performance of reasoning across different models.

| Method | Reasoning-Answer Consistency | Reasoning Content Repetitiveness | Logical Coherence & Knowledge | Relevance to Video Content |
|---|---|---|---|---|
| QWEN2.5-VL | 0.69 | **8.87** | 6.97 | 6.82 |
| VIDEO-R1 | 0.78 | 4.14 | 6.87 | 6.57 |
| MOSS-CHATV-NO-PRR | 0.72 | 7.80 | **7.80** | **7.45** |
| MOSS-CHATV | **0.79** | 7.23 | 7.59 | 7.35 |

To investigate the quality of video reasoning texts, we employed GPT-4o as a judge to conduct a multi-dimensional quality assessment of the reasoning and answers generated by models. This assessment framework comprises four core metrics, for which GPT-4o assigns a score for each dimension (detailed dimension and prompts can be found in Appendix B.1).

Process supervision within reinforcement fine-tuning demonstrates a significant contribution to enhancing the quality of reasoning across multiple dimensions. MOSS-ChatV exhibits a well-

balanced and overall excellent performance profile, shown in table 4. Compared to Video-R1, while achieving comparable performance in Reasoning-Answer Consistency, MOSS-ChatV demonstrates higher information density (*i.e.*, lower repetitiveness), more robust logical coherence, and greater relevance to video content. Furthermore, when contrasted with its variant MOSS-ChatV-no-PRR, MOSS-ChatV achieves a higher degree of Reasoning-Answer Consistency. This suggests that process supervision effectively guides the model towards generating more credible and trustworthy outputs. Although Qwen2.5-VL records the highest information density, its comparatively lower scores on other metrics imply that this conciseness might stem from unconstrained cognitive divergence, which could be detrimental to the generation of high-quality reasoning content.

## 5 RELATED WORK

### 5.1 ADVANCED VIDEO-LLM

With the burgeoning development of Multimodal Large Language Models (MLLMs), such as Qwen (Wang et al., 2024a) and InternVL (Wang et al., 2024b; 2025c), video understanding has emerged as a critical dimension for evaluating model capabilities. To enhance models' comprehension of video content, researchers have employed a variety of strategies. For instance, VideoChat-GPT (Maaz et al., 2023) focuses on improving model proficiency in video dialogue, description, and reasoning by introducing video-specific instruction-tuning datasets and a quantitative evaluation framework. Other approaches, exemplified by models like NVILA (Liu et al., 2024b), LongVU (Shen et al., 2024), and VideoLLaMA3 (Zhang et al., 2025a), enhance their capacity to process long videos through various visual token compression techniques, such as removing redundant tokens or employing MLP-based compression. Furthermore, models such as LLaVA-OV (Li et al., 2024a) are typically pre-trained on large-scale video-text pair datasets (video training data) and subsequently fine-tuned using instruction data for tasks like video question answering and description generation to adapt to diverse video understanding scenarios. These works collectively provide an excellent foundation for advancing video reasoning capabilities in Video-LLMs. While these methods advance general video understanding, our work introduces a reinforcement learning framework to directly supervise the temporal reasoning process.

### 5.2 REASONING AND REINFORCEMENT LEARNING IN VIDEO-LLMs

To enhance the reasoning capabilities of video models, researchers have made numerous attempts, such as utilizing rationale construction, structural reasoning, objective granularity, and other methods (Wang et al., 2025a). Recent advances in Reinforcement Learning (RL) have significantly improved LLM alignment and specialized capabilities, as seen in reasoning LLMs (DeepSeek-AI et al., 2025). This success has spurred RL-based enhancements for Multimodal LLMs (Yang et al., 2025; Meng et al., 2025). Specifically, for video modality, Videochat- R1 (Li et al., 2025) and TimeZero (Wang et al., 2025b) leverage RL rewards for temporal grounding, while TinyLLaVA-Video-R1 (Zhang et al., 2025b) demonstrates RL's effectiveness even on small models. Video-R1 (Feng et al., 2025) employs contrastive RL to improve temporal understanding. Our approach is distinguished by a novel, rule-based Process Reasoning Reward (PRR), which offers more granular supervision on the reasoning path itself.

## 6 CONCLUSION

Through analyzing the relationship between video state prediction tasks and video reasoning capabilities, we demonstrate their mutual reinforcement. Based on this insight, we introduce MOSS-Video, a dedicated dataset for training and evaluating video state prediction task. For reinforcement fine-tuning of video modalities, we propose Process Reasoning Reward (PRR), a rule-based reward mechanism. Comparative and ablation experiments confirm the effectiveness of our approach. Using single-task training data alone, we achieve holistic improvements in video analysis performance while maintaining stable reasoning quality. In summary, we find that model reasoning capability in video contexts deserves greater attention. Through MOSS-ChatV, we verify that reinforcement fine-tuning with process supervision significantly enhances video reasoning performance, achieving performance gains and state-of-the-art results even under low-quality video inputs.

# 7 ETHICS STATEMENT

This research complies with ethical standards. It utilizes datasets that are either synthetic or publicly available, and contains no sensitive or personally identifiable information. The study involves no direct human subjects, nor does it pose any privacy or security concerns. All methodologies and experiments were conducted in accordance with applicable laws and established research integrity practices. There are no conflicts of interest, no undue influence from external sponsorship, and no concerns related to discrimination, bias, or fairness. Moreover, this research does not lead to any harmful insights or applications.

# 8 REPRODUCIBILITY STATEMENT

We have taken steps to ensure the reproducibility of the results presented in this paper. The experimental settings, including datasets and model designs, are thoroughly described in Section 4. Source code will be made publicly available upon acceptance.

# 9 LLM USAGE STATEMENT

In this work, large language models (LLMs) were used exclusively to assist with writing, editing, and LaTeX formatting. Their role was confined to enhancing clarity, grammar, and overall presentation; they had no impact on the design of experiments, data processing, analysis, or the interpretation of results.

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

## A DETAILS OF MOSS-VIDEO

We leverage high-quality ShareGPT4Video as our primary data source, employing two parallel annotation pipelines to capture both coarse- and fine-grained object states. In the coarse-grained pipeline, GPT4-o processes each video's annotation file to produce triplets of the form ⟨Object, State, Timestamp⟩, thereby characterizing an object's state over a defined temporal interval. Concurrently, in the fine-grained pipeline, GPT4-o extracts more detailed triplets ⟨Object, State: Description, Timestamp⟩, which enrich each state with a specific textual description at a precise moment. Finally, we again invoke GPT4-o to integrate these two annotation streams into a unified temporal model of object dynamics, from which we automatically generate question–answer pairs that probe the predicted future states of objects.

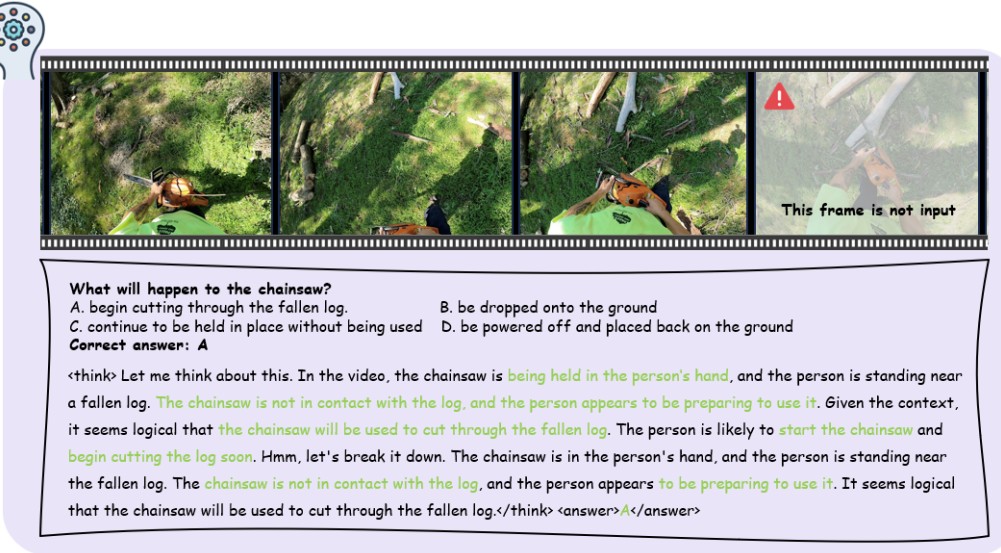

Figure A.1: The example of MOSS-Video.

## B MLLM-AS-A-JUDGE FOR REASPONSE QUALITY

We used GPT-4-o to verify the reasoning process and the final response results of the reasoning model. The specific evaluated dimensions are listed below:

**Reasoning-Answer Consistency (0 or 1)**: This is a binary metric. A score of 1 is awarded if the final conclusion of the reasoning aligns with the content of the model-selected option; otherwise, it receives a score of 0.

**Reasoning Content Repetitiveness (0-10)**: This assesses the presence of redundant information in the reasoning process. Higher repetitiveness results in a lower score, aiming to measure information density and avoid the amplification of potential biases or errors.

**Logical Coherence and Knowledge Accuracy (0-10)**: This directly evaluates the intrinsic quality of the reasoning process. The more rigorous the logic and the more accurate the application of world knowledge, the higher the score.

**Reasoning-Video Content Relevance (0-10)**: This measures how closely the reasoning is based on the video content. Higher relevance yields a higher score, aiming to penalize unfounded speculations or associations unrelated to the video.

In addition, the specificprompts used can refer to B.1.

---

**Prompts for Reasoning Process Evaluation**

Given the following video captions, extract object-centric information for a video prediction task.

**Reference Video Segments (what can be seen):**

```
for caption in reference_captions:
    prompt += f"Time {caption['time_stamp']}s:
    {caption['content']}\n\n"
```

**Prediction Segment (what needs to be predicted):**

```
Time {prediction_caption['time_stamp']}s:
{prediction_caption['content']}
```

**Task 1:** For each object in the video, provide coarse-grained information as <object, state, time> triplets for both reference and prediction segments.

**Task 2:** For each object, provide fine-grained information as <object, change description, time> triplets for how objects change over time.

**Format your response as follows:**
COARSE-GRAINED INFORMATION:
Object 1:

- At time [timestamp]: [state description]

- At time [timestamp]: [state description]

- …

Object 2:

- At time [timestamp]: [state description]

- …

FINE-GRAINED INFORMATION:
Object 1:

- From time [start] to [end]: [detailed change description]

- …

Object 2:

- From time [start] to [end]: [detailed change description]

- …

Focus only on objects that appear in the prediction segment. Be specific and detailed in your descriptions.

---

Figure A.2: Prompt template used for evaluating the reasoning process of video question answering models.

---

**Prompts for Reasoning Process Evaluation**

---

You are a professional video question answering reasoning process evaluator. Your task is to evaluate the quality of the reasoning process **ONLY**, based on the provided video frames, question, options, and a model's reasoning text. You **DO NOT** need to judge the correctness of the model's final answer.

Please evaluate the reasoning process based on the following dimensions:

**1. Reasoning Conclusion and Answer Tag Consistency (0 or 1 point):**
Criterion: Check whether the conclusion in the reasoning text semantically matches the option marked by the `<answer>` tag. You should carefully analyze and consider the logic within the `<think>` tag.
Scoring:
1 point: Consistent.
0 points: Inconsistent.

**2. Reasoning Content Repetitiveness (0–10 points, lower score for more repetition):**
Criterion: Assess whether the reasoning content contains unnecessary repetition of words, phrases, or semantics.
Scoring Guidelines:
9–10 points: Very concise, no unnecessary repetition, high information density.
6–8 points: Slight repetition or reasonable restatement for emphasis, overall flow is smooth.
3–5 points: Obvious repetition, but core idea is still discernible.
0–2 points: Massive repetition, almost no new information.

**3. Reasoning Logical Coherence and Knowledge Accuracy (0–10 points):**
Criterion: Evaluate if the reasoning steps are clear and coherent, the logical chain complete, and any assumptions reasonable and correct.
Scoring Guidelines:
9–10 points: Rigorous logic, well-organized, sufficient argumentation, accurate assumptions.
6–8 points: Generally coherent with minor flaws.
3–5 points: Obvious breaks or minor errors not affecting main conclusion.
0–2 points: Chaotic or contradictory logic, erroneous assumptions.

**4. Reasoning and Video Content Relevance (0–10 points, lower score for more deviation):**
Criterion: Assess whether observations and conclusions are strictly based on provided video frames.
Scoring Guidelines:
9–10 points: Strictly based on video content with strong evidence.
6–8 points: Primarily based on content with minor reasonable inference.
3–5 points: Mostly imagination or misunderstanding of video.
0–2 points: Completely unrelated or speculative.

**[Input Information]**

Video Frames: {num_frames_provided} frames are provided. (Actual frames are sent as image data)
Question: {question_text}
Model Reasoning Text: {model_reasoning_text}
Model's Answer Tag Content: `<answer>{model_answer_tag_content}</answer>`

**[Your Evaluation Output]**

Please provide your evaluation scores strictly in the following format, one line per dimension, containing only the score:

```
Dimension1_Score:  [0 or 1]
Dimension2_Score:  [0-10]
Dimension3_Score:  [0-10]
Dimension4_Score:  [0-10]
```

Figure B.1: Prompt template used for evaluating the reasoning process of video question answering models.

