# OpenReview forum: "MOSS-ChatV: Reinforcement Learning with Process Reasoning Reward for Video Temporal Reasoning"
_ICLR.cc/2026/Conference — ICLR 2026 Conference Withdrawn Submission_

### Official Review · Reviewer_TUFj · 2025-10-27

**Soundness:** 3
**Presentation:** 3
**Contribution:** 3
**Rating:** 4
**Confidence:** 3

**Summary:**

The paper addresses process inconsistency in video reasoning by proposing MOSS-ChatV, a reinforcement learning framework that uses DTW-based process rewards to align generated reasoning steps with ground truth annotations. Unlike methods requiring auxiliary reward models, their approach uses rule-based rewards to match reasoning traces to temporal dynamics in videos. The authors construct MOSS-Video, a benchmark with GPT-4o-annotated reasoning traces, using the training split for RL fine-tuning on models like Qwen2.5-VL and Phi2. MOSS-ChatV achieves 87.2% on MOSS-Video (test) and shows improvements on MVBench and MMVU. The framework generalizes across architectures and produces more consistent reasoning traces according to GPT-4o-based evaluation.

**Strengths:**

Writing is clear

Introduce a high quality dataset with reference reasoning trace

Design a rule-based reward that incorporate process level by alignment of the substeps.

**Weaknesses:**

The improvement is very little on the other benchmarks besides the author's own benchmark, so it would be beneficial to provide some analysis why this happen.

The idea of RL with process reward is not novel.

Given you already have reference trajectory, may be leverage a off-the-shelf LLM to directly compare the reference vs generated sequence is a good reward baseline?

**Questions:**

See weaknesses

---

### Official Review · Reviewer_hZsL · 2025-10-27

**Soundness:** 2
**Presentation:** 2
**Contribution:** 2
**Rating:** 2
**Confidence:** 4

**Summary:**

This paper addresses a challenge in the field of video reasoning for MLLMs. It identifies an important problem: "process inconsistency," where models produce correct final answers despite flawed intermediate reasoning. The paper proposes rule-based Process Reasoning Reward, which uses Subsequence Dynamic Time Warping to align generated reasoning with reference traces. This approach cleverly avoids the need for a separate, expensive-to-train reward model.

**Strengths:**

1. The paper introduces MOSS-Video, a video-state-prediction dataset with annotated reasoning traces, comprising 11.6k training and 2.8k test samples, and aims to advance process supervision in video models.
2. This paper proposes Process Reasoning Reward, motivates Subsequence DTW over naive DTW—avoiding reward hacking by rewarding the gold reasoning path within longer, exploratory outputs—and uses a ROUGE-based rule reward for simplicity, efficiency, and training stability.

**Weaknesses:**

1. Lack of Generalization and Potential Overfitting: My primary concern is the generalizability of the learned reasoning skill. While the model achieves a massive and impressive $\approx$19.6-point gain on the in-domain MOSS-Video test set (86.62% vs. 67.00% for the baseline), this improvement fails to translate to out-of-domain (OOD) benchmarks: (a) On general video understanding tasks, the gains are marginal: +0.5% on MVBench and +0.3% on VideoMME. (b) Performance on several key reasoning benchmarks significantly degrades. As shown in Table 2b, the model's score drops dramatically on VideoMMMU (52.3 $\rightarrow$ 50.2), VCR-B (38.4 $\rightarrow$ 35.3), VSI-Bmc (35.9 $\rightarrow$ 35.2) and VSI-Breg (39.7 $\rightarrow$ 28.2). Additionally, it got a marginal improvement (0.1) on RTV-Bench. This pattern—strong in-domain performance coupled with marginal gains and significant OOD degradation—strongly suggests the model is not learning a general-purpose temporal reasoning capability. Instead, it appears to be overfitting to the specific task structure and linguistic style of the MOSS-Video dataset.
2. Reward Function Penalizes Valid, Alternative Reasoning: The paper's core contribution, the Process Reasoning Reward (PRR), seems designed to enforce syntactic mimicry rather than semantic correctness, which likely causes the overfitting mentioned above. The reference reasoning traces are generated by GPT-4o , which, while practical, means the "gold standard" represents only one possible valid reasoning path, not all of them. The model-generated path is then compared to this single reference path using a flawed distance metric based on ROUGE-1, ROUGE-2, and ROUGE-L. ROUGE is a metric of N-gram overlap, not semantic equivalence. Consequently, if the model generates a reasoning step that is semantically correct but phrased differently (e.g., "The surfer is crouching to maintain balance" vs. "The surfer's low stance helps with stability"), it would receive a high distance score (low similarity) and thus a low reward. This actively penalizes correct, diverse reasoning and forces the model to learn the specific phrasing of GPT-4o, which is not a generalizable skill.
3. To strengthen the paper, several critical ablation studies on the PRR and SDTW design are needed, as the current justification is insufficient. The paper provides a qualitative argument for SDTW over naive DTW (Figure 4), but a quantitative ablation is missing. The authors should add a row to Table 3 showing the performance of "MOSS-ChatV w/ naive DTW" to empirically prove that SDTW is superior and that naive DTW leads to reward hacking. Furthermore, a key ablation is needed to compare the N-gram-based ROUGE metric against a semantic similarity metric (e.g., BERTScore or cosine similarity of sentence embeddings) to test if a semantically-aware reward could solve the "alternative phrasing" problem and improve generalization. Finally, the reward scaling hyperparameter $\alpha$ in the distance-to-reward transformation is critical but its selection and sensitivity are never discussed.

**Questions:**

See Weaknesses.

---

### Official Review · Reviewer_PBby · 2025-10-29

**Soundness:** 2
**Presentation:** 2
**Contribution:** 2
**Rating:** 4
**Confidence:** 3

**Summary:**

This paper proposes a process reward mechanism to enhance the reasoning quality of multimodal models, supported by the introduction of a new dataset, MOSS-Video. The reward aims to improve reasoning supervision through serialized text-based reasoning steps. However, the implementation of the process reward appears unreliable — the text-splitting strategy may distort original contextual information. Furthermore, the contribution of the MOSS-Video dataset is insufficiently demonstrated: the dataset’s quality, especially in the testing set, is not verified, and there is a lack of experiments showing that training on this dataset yields generalizable improvements. The paper also lacks qualitative analyses. Overall, the model underperforms compared to Video-R1 on several out-of-distribution (OOD) reasoning benchmarks (e.g., VSI-Bench, VCRBench), raising concerns about the reliability and generalizability of the proposed method.

**Strengths:**

1. The authors conduct evaluations on multiple benchmarks and report improvements on the MOSS-Video test set, though the generalization remains limited.

2. The paper explores the idea of improving reasoning quality through a rule-based process reward, which is an interesting direction despite limited empirical validation.

3. This work introduces a dataset for both training and evaluation.

**Weaknesses:**

1. The Reasoning Step Serialization process is underexplained. It is unclear how the segmentation preserves the semantics of the original reasoning. The claim that it “enables finer-grained analysis without affecting overall temporal information” lacks empirical or theoretical support.

2. The MOSS-Video dataset construction process is insufficiently detailed in the main paper, with no evidence of quality verification for the testing set.

3. The ablation experiments are limited to general understanding benchmarks (VideoMME, MVBench, and MOSS-Video) and do not demonstrate clear gains. The effectiveness of the dataset remains unproven — for instance, how would the method perform if trained on the Video-R1 dataset?

4. The reasoning quality evaluation relies solely on an MLLM-as-a-judge approach, which lacks human evaluation or inter-rater validation to ensure consistency and correctness.

5. Why not evaluate the reasoning quality on other reasoning benchmarks: like VCRBench?

6. No qualitative analyses.

**Questions:**

Please refer to the **Weaknesses**

---

### Official Review · Reviewer_qTXK · 2025-11-01

**Soundness:** 3
**Presentation:** 2
**Contribution:** 3
**Rating:** 2
**Confidence:** 3

**Summary:**

The paper introduces MOSS-ChatV, a RL framework designed to improve temporal reasoning in MLLMs for videos. Current MLLMs have low interpretability and robustness due to inconsistent or illogical intermediate reasoning. MOSS-ChatV addresses this with a new Process Reasoning Reward (PRR) that enforces temporal and logical coherence during training. Evaluation results reveal that the proposed MOSS-ChatV achieves better performance on many benchmark compared with existing baselines.

**Strengths:**

1. This paper considers a novel idea in improving MLLM's performance, which is through improving the accuracy of reasoning process. The proposed Process Reasoning Reward is novel.
2. This paper provides a dataset called MOSS-Video with reasoning traces, which can help the community better evaluate the reasoning process.

**Weaknesses:**

1. The performance improvement over Video-R1 on reasoning benchmark (as shown in Table 2) is not significant. Video-R1 got 49.40 on average, while MOSS-ChatV got 49.75 on average, which is only 0.35 improvement. In addition, both MOSS-ChatV and Video-R1 receives three SOTA among those benchmark, which further limiting the improvement. If we don't consider the proposed benchmark MOSS-Video (due to potential bias and over-fitting) and only consider well-known public benchmark, the average performance is 1.8 lower than Video-R1.
2. The use of ROUGE score will lead to inaccurate result. When using ROUGE score, a model could receive high reward for word-level similarity while still misunderstanding the underlying event or logic.
3. The dataset is generated via GPT-4o, which introduces potential bias or inconsistency. Also, there's no human annotator ensuring the accuracy of the generated dataset.
4. The submission is not using the correct template. The header "Under review as a conference paper at ICLR 2026" is not present.

**Questions:**

1. Could you please explain the prompt you used for Figure 1? Qwen2.5-VL and GPT-4o is not reasoning models, and will not generate output with format like "<think>" by default.
2. What's the motivation behind the use of ROUGE score? There are lots of alternatives (e.g. sentence-transformers) with better accuracy.

---

### Note · Authors · 2025-11-27

I have read and agree with the venue's withdrawal policy on behalf of myself and my co-authors.